# Pancreatic Cystic Tumors: A Single-Center Observational Study

**DOI:** 10.3390/medicina59020241

**Published:** 2023-01-27

**Authors:** Beata Jabłońska, Arkadiusz Gudz, Tomasz Hinborch, Bartosz Bujała, Katarzyna Biskup, Sławomir Mrowiec

**Affiliations:** 1Department of Digestive Tract Surgery, Medical University of Silesia, Medyków 14 St., 40-752 Katowice, Poland; 2Student Scientific Society, Department of Digestive Tract Surgery, Medical University of Silesia, Medyków 14 St., 40-752 Katowice, Poland

**Keywords:** pancreatic cystic tumor, intraductal papillary mucinous neoplasm, mucinous cystic neoplasm, serous cystic neoplasm, solid pseudopapillary neoplasm, cystic pancreatic neuroendocrine neoplasm

## Abstract

*Background and Objectives*: The aim of the study was to analyze the prevalence and characteristics of pancreatic cystic tumors (PCTs). *Material and Methods*: A retrospective analysis of the medical records of 124 patients, 102 (69%) women and 46 (31%) men, who had undergone surgery for pancreatic cystic tumors in 2014–2018. Among 148 pancreatic cysts, 24 (16%) were non-neoplasmatic and 124 (84%) were neoplasmatic. The neoplasmatic cysts (*n* = 124) were included in our analysis. There were five main types of PCTs: IPMN (intraductal papillary mucinous neoplasm) (*n* = 45), MCN (mucinous cystic neoplasm) (*n* = 30), SCN (serous cystic neoplasm) (*n* = 28), SPN (solid pseudopapillary neoplasm) (*n* = 8), and CPEN (cystic pancreatic endocrine neoplasm) (*n* = 8), as well as mixed-type tumors (*n* = 5). *Results:* A statistically significant dependency between PCT type and age was proven (*p*= 0.0001): IPMNs were observed in the older group of patients with an average age of 66.12 (40–79) years while SPNs were noted in the youngest group of patients with an average age of 36.22 (22–55) years. A statistically significant association between PCT type and gender (*p* = 0.0001) was found: IPMNs occurred among 24 (53.33%) men and 21 (46.6%) women. In the MCN and SPN groups, all patients were female (100%). Among the SCN group, the majority were women (27 (96.43%)), and there was only 1 (3.57%) man. A statistically significant dependency between PCT type and size was proven (*p* = 0.0007). The mean size of IPMNs was the smallest 2.95 (0.6–10 cm) and the mean size of MCNs was the largest 6.78 (1.5–19 cm). A statistically significant dependency between PCT type and tumor location was proven (*p* = 0.000238). The most frequent location of IPMN was the pancreatic head: 27 (60%). MCN was most frequently located in the pancreatic tail (18 (60%)). Most (10/28) SCNs were found in the pancreatic tail (10 (35.71%)). CPENs were most frequently located in the pancreatic tail (three (37.5%)) and pancreatic body and tail (three (37.5%)). SPNs were located commonly in the pancreatic head (five (62.5%)). The type of surgery depended on the tumor location. The most frequent surgery for IPMNs was pancreatoduodenectomy (44.4%), while for MCNs and SCNs, it was distal pancreatectomy (81%). The postoperative morbidity and mortality were 34.68% and 1.61%, respectively. Postoperative pancreatic fistula (POPF) was the most frequent (29%) complication. *Conclusions*: IPMN was the most frequent resected PCT in our material. A statistically significant association between the type of cyst and location within the pancreas, size, local lymph node involvement, and patient’s age and sex was proved. POPF was the most frequent postoperative complication. In patients with PCTs, due to substantial postoperative morbidity, adequate patient selection, considering both the surgical risk as well as the long-term risk of malignant transformation, is very important during qualification for surgery.

## 1. Introduction

Pancreatic cysts are divided into pseudocysts (75%) and true cysts (25%) [1]. Among true cysts, we distinguish neoplasmatic (10%), congestive (10%), and congenital (5%) cysts. Cystic pancreatic neoplasms account for 1–5% of all primary pancreatic tumors [1]. There are five main types of pancreatic cystic tumors (PCTs): IPMN (intraductal papillary mucinous neoplasm), MCN (mucinous cystic neoplasm), SCN (serous cystic neoplasm), SPN (solid pseudopapillary neoplasm), and CPEN (cystic pancreatic endocrine neoplasm) [2]. Due to the differences in their malignancy potential, they require precise imaging: ultrasound (US), computed tomography (CT), magnetic resonance imaging (MRI), and endoscopic ultrasound (EUS) with biopsy. A detailed view of tumor allows providing proper treatment [3,4]. It should also be mentioned that cysts are often diagnosed accidentally in imaging examinations (US, CT, and MRI). According to Chang YR et al., even 2.1% of the healthy population may have asymptomatic PCT [5]. The recent advancement in the diagnostic imaging of gastrointestinal diseases including CT and MRI has caused an increase in the incidence of recognized PCTs. The growing number of performed imaging examinations is a source of many findings that require a surgeon’s consultation. There are some different guidelines regarding the indications for surgery in patients with PCTs. Not all patients require surgical treatment. Some patients with PCTs may be carefully observed. A decision on surgical treatment depends on clinical manifestation (jaundice, new-onset diabetes, and pancreatitis), tumor type, size and morphology (mural nodules), and serum level of CA 19.9 [2,6,7].

The largest number of PCT analyses comes from Asian countries. In this region, gastrointestinal cancers represent a significant proportion of all cancers; hence, the epidemiological situation, including PCT, is well-documented. It should be taken into account that the different geographical incidence of PCTs, as well as the different distribution of their types, does not allow adapting the results of large Asian analyses to the epidemiological situation in Polish society. An example is a large analysis carried out in South Korea, in which the main type of PCT was IPMN (82.3% of cases), then MCN (only 1.5%), which is not consistent with the main Polish authors [8,9].

The diagnostics of PCTs are still very difficult and challenging because it is very important to differentiate PCTs from benign postinflammatory pancreatic cysts. Additionally, the differential diagnosis of PCT types is very essential because not all PCTs require surgery. Taking into account perioperative morbidity following pancreatic surgery as well as a higher risk of cancerous transformation in some PCTs, a proper qualification for surgical treatment is crucial in order to avoid non-necessary surgery and not to delay surgical treatment if needed. Therefore, proper preoperative diagnostics to describe a PCT type are crucial for management [10].

Currently, clinical assessment and radiological investigations (computed tomography (CT), magnetic resonance imaging (MRI), magnetic resonance cholangiopancreatography (MRCP), as well as endoscopic ultrasonography (EUS) with fluid analysis (fluid cytology, carcinoembryonic antigen (CEA) level, and biopsy of the cystic wall)) are used in the standard diagnostic process in patients with PCTs [11,12,13]. The differentiation of benign, premalignant, and malignant PCTs is important because of the difference in their treatment. EUS is used in patients in whom high-risk stigmata or worrisome features are present in a PCT and in whom its results impact on management and indications for surgery [13]. According to the literature, a cytological investigation of the cyst fluid can differentiate mucinous from nonmucinous PCTs with 42% sensitivity and 99% specificity, while cyst fluid CEA ≥ 192 ng/mL can differentiate mucinous from nonmucinous cysts with 52–78% sensitivity and 63–91% specificity. Cyst amylase level <250 U/L (4.2 μkat/L) can exclude the presence of a pseudocyst with 44% sensitivity and 98% specificity. Fluid cytology and CEA level are not useful investigations in the differentiation of mucinous PCTs such as MCN and IPMN. EUS-guided needle-based confocal laser-scanning endomicroscopy (CLE) is useful for the visualization and assessment of the pancreatic cyst epithelial wall [11]. This investigation, coupled with intravascularly injected fluorescein dye, shows the vascular PCT pattern revealing distinct epithelial features [12]. Needle-based CLE used with EUS-based fine-needle aspiration and/or biopsy might improve diagnostic accuracy in order to reduce unnecessary surgery and patient observation [11]. A multidisciplinary approach with a combination of analysis of medical history and clinical presentation, as well as the above-mentioned investigations of cross-sectional radiological imaging (CT, MRI, and MRCP), EUS with cyst fluid cytology, cyst fluid analysis, and cyst wall biopsy with CLE and molecular profiling, is used in differential diagnostics and risk stratification of PCTs [12].

The aim of the study was to assess the epidemiological situation of PCTs in the Department of Digestive Tract Surgery, a high-volume Polish pancreatic center. We also collected data about the surgical treatment of these tumors.

## 2. Materials and Methods

We retrospectively reviewed the medical records, operation protocols, and histopathological results of patients who underwent surgery for PCTs in 2014–2018 in the Department of Digestive Tract Surgery of Medical University of Silesia, Katowice, Poland. In our study, the patients were qualified for surgery according to the European Study Group guidelines. Following these guidelines, we distinguished patients with absolute indications for surgery in whom there was no need for further investigation and those who did not have significant comorbidities but who manifested one or more relative indications—in these groups, surgical treatment was advised. Absolute indications consisted of jaundice, main-pancreatic-duct dilation >10 mm, and mural nodule >5 mm. Relative indications included: pancreatitis, main pancreatic duct 5–10 mm, mural nodule <5 mm, cyst size >4 cm, CA 19–9 level, and new-onset diabetes. These guidelines stand out from others in that they contain new-onset diabetes and a cyst size cut-off of 4 cm [6]. Ultrasound (US) and computed tomography (CT) of the abdominal cavity were performed in all patients before surgery. Qualification for surgery was estimated based on CT findings. In some patients with doubtful PCTs, magnetic resonance imaging (MRI) and endoscopic ultrasonography (EUS) with biopsy were carried out. CT scans of the main types of PCTs are presented in Figure 1A–D.

Our analysis included 124 (84% of all pancreatic cysts) neoplasmatic cysts. There were 5 main types of PCTs: IPMN (*n* = 45), MCN (*n* = 30), SCN (*n* = 28), SPN (*n* = 8), CPEN (*n* = 8), and mixed types (*n* = 5). We analyzed associations between the cyst type and: age, sex, body mass index (BMI), location, size, metastatic activity, and smoking.

The Statistica 12.0 program (StatSoft Poland, Kraków, Poland) was used for statistical analysis. The normality of the distribution was assessed using the Shapiro–Wilk test. The non-parametric Kruskal–Wallis ANOVA tests were used for quantitative variables, while the chi-square test was used for qualitative variables. Then, a post hoc study was carried out, comparing individual groups among themselves. A 95% confidence level was assumed in all tests, and *p* < 0.05 was considered statistically significant.

## 3. Results

### 3.1. Characteristics of Pancreatic Cystic Tumors

A statistically significant dependency between PCT type and age was proven (*p*= 0.0001) (ANOVA rank Kruskal–Wallis test and post hoc rank Kruskal–Wallis test). IPMNs were observed in the older group of patients with an average age of 66.12 (40–79) years while SPNs were noted in the youngest group of patients with an average age of 36.22 (22–55) years. The age distribution in all patient groups is presented in Table 1.

A statistically significant association between PCT type and gender (*p* = 0.0001) was found (chi-square test). IPMNs occurred among 24 (53.33%) men and 21 (46.6%) women. In the MCN and SPN groups, all patients were female (100%). Among the serous cystadenoma (SCA) group, the majority were women (27 (96.43%)), and only 1 (3.57%) was a man. There were five women (62.5%) and three men (37.5%) in the CPEN group. The sex distribution in all main PCT types is presented in Figure 2. Statistical analyses are presented in Table 2.

A statistically significant dependency between PCT type and size was proven (*p* = 0.0007) (ANOVA rank Kruskal–Wallis test and post hoc rank Kruskal–Wallis test). The mean size of IPMNs was the smallest at 2.95 (0.6–10 cm) and the mean size of MCNs was the largest at 6.78 (1.5–19 cm). The comparison of sizes between all PCT types is presented in Figure 3. All the sizes of PCTs are presented in Table 3.

A statistically significant dependency between PCT type and tumor location was proven (*p* = 0.000238) (chi-square test). The most frequent location of IPMN was pancreatic head (27 (60%)). MCN was most frequently located in the pancreatic tail (18 (60%)). Most (10/28) SCNs were found commonly in the pancreatic tail (10 (35.71%)). CPEN was most frequently located in the pancreatic tail (3 (37.5%)) and the pancreatic body and tail (3 (37.5%)). SPN was located commonly in the pancreatic head (5 (62.5%)). A comparison of location distribution between PCTs is presented in Figure 4. All location distributions are presented in Table 4.

A statistically significant association between PCT type and malignant transformation as well as local lymph node metastases was observed in our analysis (*p* = 0.00505) (chi-square test). There were: 10 (22%) local lymph metastases in the IPMN group, 2 (7%) in the MCN group, and 2 (25%) in the CPEN group (Figure 5). Only one distant metastasis was observed in the MCN group (0.8% of all PCTs). There were no metastases in the SCN and SPN groups.

We did not prove a statistically significant relationship between the type of PCT and patients’ BMI (*p* = 0.158). Additionally, dependency between the type of PCT and smoking was not proved (*p* = 0.431).

### 3.2. Surgical Details and Postoperative Complications

The most frequent surgery among IPMN was pancreatoduodenectomy (PD) (44.4%), while among MCN and SCN, it was distal pancreatectomy (DP) (81%). The type of surgery depended on tumor location. There was no difference in PD or DP outcomes or in complications, depending on the type of PCT. There was association between surgical procedure and operation time (*p* < 0.0001). Postsurgery hospitalization time was longer in patients following PD (Figure 6). The complete presentation of surgery types is visible in Table 5.

Postoperative morbidity and mortality were 34.68% and 1.66%, respectively. Intra-abdominal fluid collection and clinically relevant postoperative pancreatic fistula (POPF) were the most frequent complications. All postoperative complications are presented in Table 6. There was an association between American Society of Anesthesiologists’ (ASA) classification and the postoperative complication rate. A higher number of complications was noted in patients with a higher ASA score (Figure 7). The Spearman test showed a significant positive correlation between the ASA score and hospitalization time in patients undergoing DP (body + tail) with splenectomy (R = 0.614; *p* = 0.00308). There was no significant correlation between the ASA score and hospitalization time in patients undergoing DP (tail) with splenectomy, PD (Traverso), and DP (body + tail). There was no significant correlation between body mass index (BMI) in patients undergoing all the above-mentioned procedures.

## 4. Discussion

IPMNs were the most common PCTs in our patients’ is most often recognized in the 6th–7th decade of life in males, which was confirmed in our analysis. This is a kind of exception because the vast majority of other PCTs are more common among women [14,15]. These data were also confirmed by our analysis. IPMNs may show different degrees of malignancy: from benign lesions with low-grade dysplasia, through cancer in situ, to invasive cancer with local and distal metastases [14]. Our results refer to IPMN in general. Sendai’s research group (2004) recognized that due to the high incidence of invasive IPMN and carcinoma in situ (approximately 70% of resected tumors), surgical resection is recommended [10]. This indicates that they are one of the most malicious PCTs with the highest potential for metastasis. According to Sugiyama et al., metastases to lymph nodes occur in about 23% of cases, which is in line with our results (22%) [15]. The most common IPMN location according to many publications is the head of the pancreas (approx. 58.1–67%). In our analysis, it was 60% of cases, which corresponds with the worldwide literature [16,17]. In our analysis, IPMN reached the smallest sizes: avg. 2.95 cm (0.6–10 cm). This may be due to the characteristic location in the pancreatic ducts, which causes the early onset of symptoms and an earlier diagnosis.

A large group of resected tumors comprised SCNs, which are treated as benign lesions requiring only observation. Unfortunately, the proper identification of SCNs based on imaging tests can be difficult—they can imitate MCN or IPMN tumors. Therefore, in doubtful cases, they are resected. The final diagnosis is given with the results of histopathological examination [18,19,20,21,22]. They are most often found among women in the 6th decade of life within the body and tail of the pancreas [23]. Our research confirms these reports. Due to the small percentage of SCN malignancy, oscillating around 3%, long-term observation is recommended. Resections are performed when symptoms or malignancy features appear in control tests, which results in a relatively large tumor diameter at the time of resection—in our study, 4.59 cm (1.9–11 cm) [24]. A feature of SCN that we could observe in our analysis is the lack of metastases to local lymph nodes. This is due to the benign nature of this PCT.

MCNs are usually large, multichamber PCTs [25]. They occurred only among middle-aged women (47.47), which is in accordance with the world literature [26]. The presence of MCN among the vast majority of women may be associated with the near location of the left primary gonad and pancreas during organogenesis. This is also evidenced by the fact that MCN histologically resembles the stroma of the ovary [26]. However, it should be mentioned that, sporadically, MCN occurs among men, especially the elderly [27]. The authors suggest that this indicates that they are hormone-sensitive neoplasms [28,29]. Among all the analyzed groups of PCTs, this type of neoplasm reached the largest average size of 6.78 cm (1.5–19 cm). The fact that MCNs have the ability to reach larger sizes compared with other PCTs has also been shown in other publications, where the average size reached over 10 cm [30]. MCNs in the analyzed group were mainly located in the tail and less often in the body of the pancreas. Our data correspond with the literature [25,28,30]. In the study group, the incidence of lymph node metastasis was 7%, which differs from other publications, where the incidence of metastases reached 33%, which is why this type of PCT is considered malignant regardless of its stage [2,3,4,25]. It should be noted that the only distant metastasis was recorded in MCN. It was located in the liver, which, according to the literature, is a common location for distant metastases in this group [26,27].

SPNs are rare PCTs accounting for about 9% of all PCTs [26]. Their rare occurrence was confirmed in our study. They constituted 6.45% of all PCTs in our patients. These asymptomatic PCTs usually are reported in young females, which was confirmed in our analysis [30,31,32]. In the analyzed group, women constituted 100% of cases, which was consistent with the literature, since 85% of SPNs occur in women [32]. The prevalence of SPN in the female population compared to men is estimated at 8–10: 1 [31,33]. The fact that SPN occurs most often in the female population may be associated with the presence of progesterone receptors (81% of cases) and estrogen receptors on this neoplasm cells [34]. The presence of only women in our group may be due to the insufficient number of analyzed patients (eight cases of SPN), whereas it indicates a trend presented in other publications. In the collected material, this type of PCT was mainly located in the head of the pancreas (62.5%), and rarely in its distal parts, which was confirmed in other reports [31,35]. The average cyst size in this group was 4.35 cm (median 3.15 cm). The occurrence of one cyst with a diameter of 14 cm affected the average size. However, in the literature, we could find SPNs larger than in our data [36]. In our study, none of the eight SPN cases showed metastases to either local or distant lymph nodes. In the analysis conducted on 553 patients with SPN, Yu et al. showed that lymph node metastases can occur in up to 10% of all PCT cases [28]. Malignant SPN cases with distant liver and peritoneal metastases are also known [36,37,38].

CPENs are the cystic subtype of neuroendocrine tumors [26,39]. The prevalence of CPENs in a given gender varies in age categories. At a younger age (20–29 years), they are the least common and it is the only group in which they occur more often among women. The incidence increases with age, reaching 15.9/1 million, 10.3 men and 5.6 women, respectively [40]. Data on the occurrence of cystic subtype depending on gender and age are not available; therefore, our results refer to CPEN in general. In the analyzed group, there were, respectively, five (62.5%) women and three (37.5) men. Such a small group does not allow drawing clear conclusions. Additionally, the age range of the examined group was 22.84–68.33 (52.56 on average), which does not allow an unequivocal statement as to the age group but suggests the occurrence of CPEN in older age. These tumors reach relatively small sizes (mean 3.44 cm) compared to other types; however, in the literature, CPEN can be found with dimensions up to 25 cm. According to the literature, they are located mainly in the body and tail (37.5%) [41]. In our analysis, they metastasized to the lymph nodes most often of all PCTs described (25%). A small group of examined cases could have affected our results. The most useful factor to differentiate the risk of malignancy is tumor size > 20 mm. According to the literature, this is a limit value indicating the appearance of the malicious nature of CPEN [42]. However, it seems that this may be a wrong assumption because, in our work, there were two cases of metastases to local lymph nodes where the tumor sizes turned out to be much smaller.

The effect of BMI on the occurrence of a specific type of PCT was also tested. Such dependency has not been proven (*p* = 0.158); however, according to Mizuno et al., the occurrence of PCTs is associated with obesity and diabetes [43]. There are also reports that, among patients with higher BMI (BMI > 25 kg/m^2^) in both MCN and IPMN types, the percentage of malignant forms is higher: for MCN (*p* = 0.0001) and IPMN (*p* = 0.018) [44].

The effects of smoking on the development of pancreatic cancer are known. In our analysis, we tried to show the relationship between smoking and the onset of a specific type of PCT. We were unable to prove this dependency (*p* = 0.431). However, there have been reports that smoking can affect the development of adenocarcinoma among patients who already have noninvasive IPMN [45].

The type of performed surgery mostly depends on the tumor’s location within the pancreas. The tumors were located mostly in the pancreatic corpus and tail; hence, the most frequent procedures were various types of distal resections. Although spleen preservation is recommended due to a lower rate of postoperative infectious complications, it is not always possible to perform it—resections with splenectomy are more common in our material [46]. In patients with the tumor located in the pancreatic head, PD is a method of choice. The most frequent reconstruction type was the Traverso method.

Taking into account the above-mentioned factors, the most frequent surgery among IPMN was PD (44.4%), while among MCN and SCN, it was DP with or without splenectomy that was associated with the location of these PCTs. In the group of MCN, there was 43.33% of DP with splenectomy and 30% of extended DP with splenectomy. The other procedures were: DP (13.33%) and extended DP (6.67%). In the SCN group, the most common procedure was DP with splenectomy (35.71%) and extended DP with splenectomy (25%). There was only one (3.33%) DP and extended DP in this group. So, the type of surgery depended on tumor location. We reported longer hospital stays in patients following PD compared to patients undergoing DP. The above-mentioned reports correspond with the literature data [47].

We compared our reports regarding PCT distribution and location as well as types of surgery and postoperative outcomes with the other largest studies in the worldwide literature. IPMN was the most frequent PCT type in our study, which was similar to Valsangkar et al.’s [48] study in which IPMNs were reported in 38% of patients. The other PCTs at the Massachusetts General Hospital were as follows: MCN (23%), SCN (16%), CPEN (7%), and SPN (3.4%). The location and surgery type distribution was also similar to our patients [48]. The similar distribution of PCT types as well as surgery types was also presented in Del Chiaro et al.’s [49] study from Sweden in which there were 51.7% of IPMNs, 23.4% of SCNs, 17.8% of MCNs, and 5.7% of CPENs [49]. The distribution of PCT location and types of surgery was different in the Indian study by Chaudhari et al. [47] in which MCN was the most frequent PCT (30.2%) followed by SPN (28.6%), SCN (23.2%), IPMN (8%), and CPEN (3.8%) [48]. In the other Eastern study performed on a large Chinese population, SPN was the most frequent PCT (31.67%) followed by SCN (30%), IPMN (22%), MCN (16.2%), and CPEN (3.8%) [50]. Therefore, the epidemiological situation regarding PCTs in the Polish population is similar to American and other European populations and different from Eastern populations. Regarding postoperative morbidity and mortality, our results (34.68% and 1.61%, respectively) were similar to the other studies. The postoperative morbidity and mortality rates were as follows: 38% and 0.5% (Valsangkar et al.) [48], 39.7% and 1.4% (Del Chiaro et al.) [49], and 28.9% and 0.9% (Chaudhari et al.) [47]. The limitation of this study is the single-center and retrospective analysis as well as the small number of CPEN (*n* = 8) and SPN (*n* = 8) groups. The single-center nature of our study limits the impact of confounding factors related to differences in the surgical technique and clinical experience for surgical outcomes in patients who received surgery in different centers. Although our study included a smaller cohort, it was a Polish/Central European voice in the discussion on this subject.

A prospective, multicenter study is needed to show the epidemiology and treatment results of PCTs in Poland. Despite the retrospective and single-center design of our study, its results can be used in a large meta-analysis regarding epidemiology and surgical outcomes in patients with PCTs. It is also possible to compare these results with the other worldwide literature data regarding PCTs.

## 5. Conclusions

In conclusion, this is the first and largest one-center report from Poland for resected PCTs. Female predominance and distal pancreatic location in most PCTs (besides IPMN with male and pancreatic head predominance), young age, and the benign nature of most PCNs were noted. IPMNs were the smallest while MCNs were the largest tumors in our patients. Malignant transformation and metastases to lymph nodes were reported in patients with CPENs, IPMNs, and MCNs. One distant metastasis was reported in a patient with MCN. There was no metastasis in patients with SCNs and SPNs. According to our analysis, the distribution of different types of PCTs is similar to American and other European populations and differs from Eastern (Indian and Chinese) publications.

Due to substantial postoperative morbidity and a risk of malignancy in patients undergoing surgery for PCTs, the proper selection of candidates for surgical treatment is very important and challenging. Management in patients with PCTs is still very challenging due to difficulties in the differentiation of nonmalignant and malignant neoplasms, which determines the proper decision. The most important is the proper selection of patients requiring surgery at the right time, without unnecessarily exposing patients who do not require surgical treatment to complications related to surgery. Our study as well as the other reports have showed that pancreatic resection performed even in high-volume surgical centers is associated with a relatively significant risk of postoperative morbidity despite progression in surgical techniques. The correct algorithm of surveillance of patients not qualified for resection is also important. Knowledge of epidemiology and clinical presentation of specific types of PCTs is as important for clinicians as modern radiological cross-sectional investigations in differential diagnostics and making decisions regarding the management of PCT. Therefore, many further retrospective and prospective and single- and multicenter studies, as well as meta-analyses based on their results, are needed in order to modify and precise current guidelines for the management of PCTs.

## Figures and Tables

**Figure 1 medicina-59-00241-f001:**
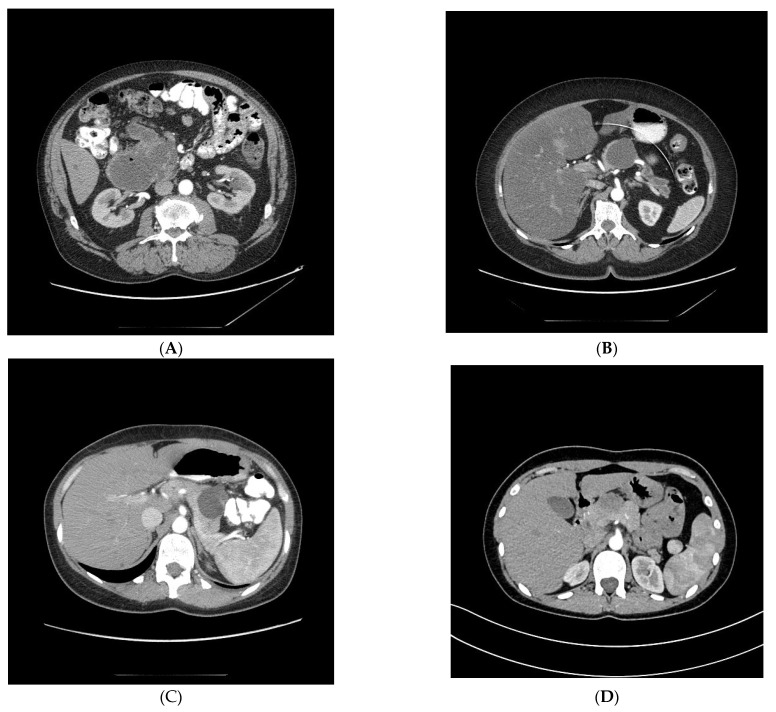
(**A**). Intraductal papillary mucinous neoplasm (IPMN) in computed tomography (CT) of the abdominal cavity. (**B**). Mucinous cystic neoplasm (MCN) in computed tomography (CT) of the abdominal cavity. (**C**). Serous cystic neoplasm (SCN) in computed tomography (CT) of the abdominal cavity. (**D**). Solid pseudopapillary neoplasm (SPN) in computed tomography (CT) of the abdominal cavity.

**Figure 2 medicina-59-00241-f002:**
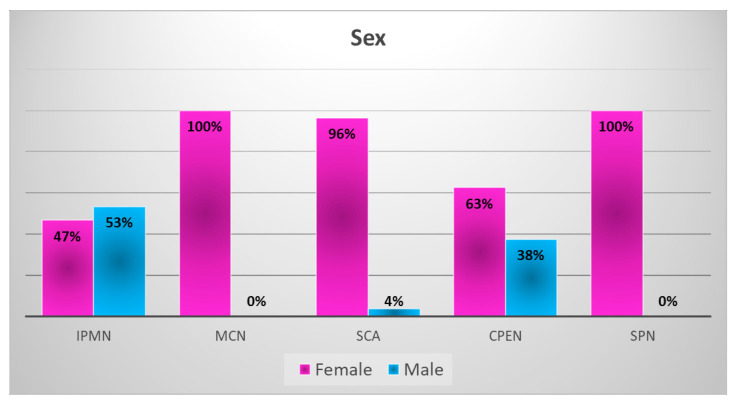
Sex distribution in main pancreatic cystic tumor types. IPMN, intraductal papillary mucinous neoplasm; MCN, mucinous cystic neoplasm; SCA, serous cystadenoma; CPEN, cystic pancreatic endocrine neoplasm; SPN, solid pseudopapillary neoplasm.

**Figure 3 medicina-59-00241-f003:**
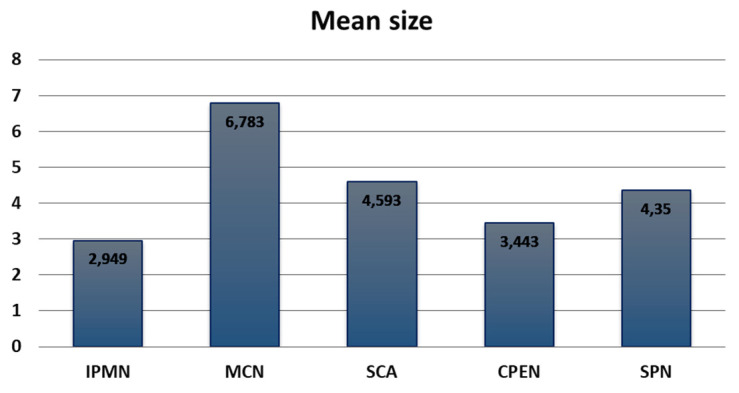
Comparison of sizes between all pancreatic cystic tumor types. IPMN, intraductal papillary mucinous neoplasm; MCN, mucinous cystic neoplasm; SCA, serous cystadenoma; CPEN, cystic pancreatic endocrine neoplasm; SPN, solid pseudopapillary neoplasm.

**Figure 4 medicina-59-00241-f004:**
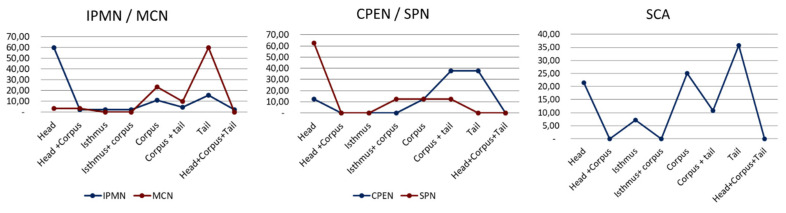
Comparison of location distribution between pancreatic cystic tumors. IPMN, intraductal papillary mucinous neoplasm; MCN, mucinous cystic neoplasm; SCA, serous cystadenoma; CPEN, cystic pancreatic endocrine neoplasm; SPN, solid pseudopapillary neoplasm.

**Figure 5 medicina-59-00241-f005:**
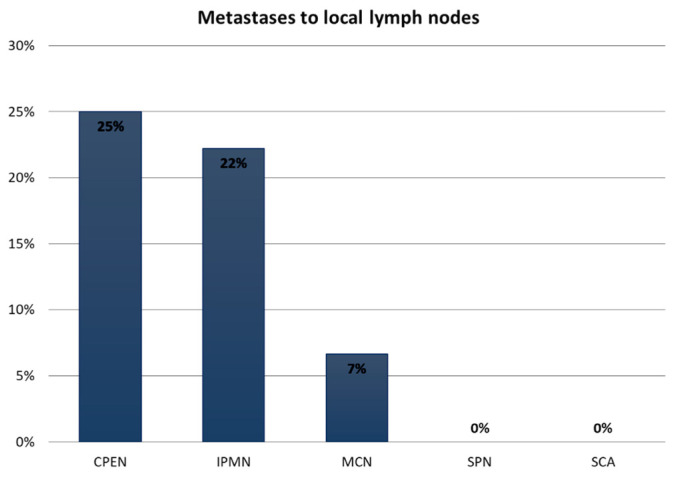
Frequency of metastases to lymph nodes in pancreatic cystic tumors. IPMN, intraductal papillary mucinous neoplasm; MCN, mucinous cystic neoplasm; SCA, serous cystadenoma; CPEN, cystic pancreatic endocrine neoplasm; SPN, solid pseudopapillary neoplasm.

**Figure 6 medicina-59-00241-f006:**
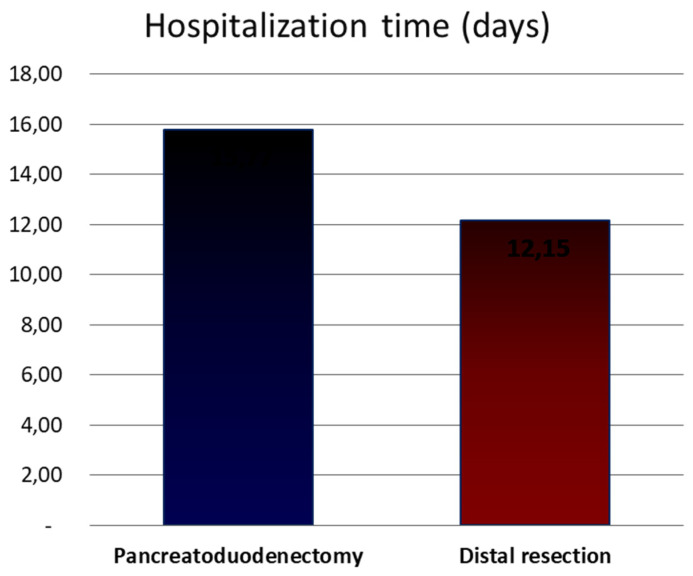
Comparison of duration of hospitalization depending on the type of surgery.

**Figure 7 medicina-59-00241-f007:**
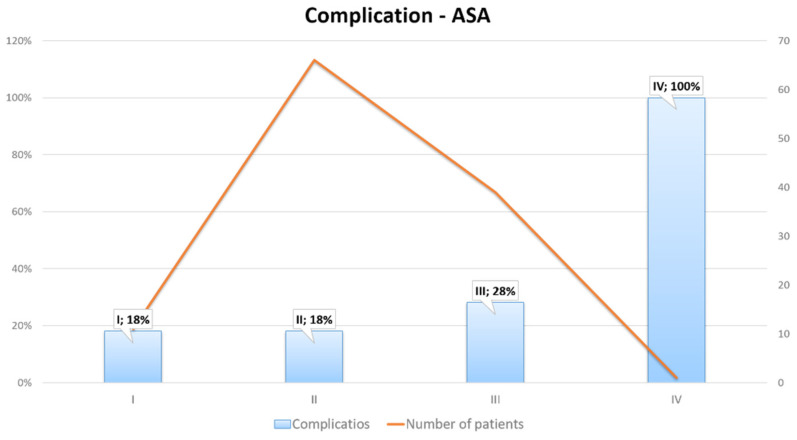
Association between ASA classification and complication rate.

**Table 1 medicina-59-00241-t001:** Type of pancreatic cystic tumors (PCTs) and age.

Group	*n*	Median	Lower Quartile	Upper Quartile	Min.	Max.
IPMN	45	66.847	62.783	70.967	40.681	79.65
MCN	30	46.714	36.755	60.64	25.228	72.353
SCN	28	62.725	53.179	68.458	29.853	78.467
CPEN	8	58.542	48.676	63.847	22.839	68.331
SPN	8	33.799	25.828	43.398	22.908	55.778

IPMN—intraductal papillary mucinous neoplasm, MCN—mucinous cystic neoplasm, SCN—serous cystic neoplasm, CPEN—cystic pancreatic endocrine neoplasm, SPN—solid pseudopapillary neoplasm.

**Table 2 medicina-59-00241-t002:** Association between pancreatic cystic tumor (PCT) types and sex.

Chi-Square Test	Female		Male		*p*
p(IPMN:MCN)	IPMN 21	MCN 30	IPMN 24	MCN 0	*p* < 0.0001
p(IPMN:SCN)	IPMN 21	SCN 27	IPMN 24	SCN 1	*p* < 0.0001
p(IPMN:SPN)	IPMN 21	SPN 8	IPMN 24	SPN 0	0.00101
p(MCN:CPEN)	MCN 30	CPEN 5	MCN 0	CPEN 3	0.00126
p(SCN:CPEN)	SCN 27	CPEN 5	SCN 1	CPEN 3	0.0151
p(CPEN:SPN)	CPEN 5	SPN 8	CPEN 3	SPN 0	0.0275

IPMN—intraductal papillary mucinous neoplasm, MCN—mucinous cystic neoplasm, SCN—serous cystic neoplasm, CPEN—cystic pancreatic endocrine neoplasm, SPN—solid pseudopapillary neoplasm.

**Table 3 medicina-59-00241-t003:** Size of different pancreatic cystic tumor (PCT) types.

		Size (cm)			
Type	*n*	Average	Min.	Max.	Standard Deviation
IPMN	45	2.95	0.6	10	1.82
MCN	30	6.78	1.5	19	4.029
SCN	28	4.59	1.9	11	2.621
CPEN	8	3.44	1.5	9	2.628
SPN	8	4.35	1.3	14	4.143

IPMN—intraductal papillary mucinous neoplasm, MCN—mucinous cystic neoplasm, SCN—serous cystic neoplasm, CPEN—cystic pancreatic endocrine tumor, SPN—solid pseudopapillary neoplasm.

**Table 4 medicina-59-00241-t004:** Localization of pancreatic cystic tumors (PCTs) within pancreas.

	IPMN (*n* = 45)	MCN (*n* = 30)	SCA (*n* = 28)	CPEN (*n* = 8)	SPN (*n* = 8)
Head	27 (60%)	1 (3.33%)	6 (21.43%)	1 (12.5%)	5 (62.5%)
Head + corpus	1 (2.22%)	1 (3.33%)	0	0	0
Isthmus	1 (2.22%)	0	3 (7.14%)	0	0
Isthmus +corpus	1 (2.22%)	0	0	0	1 (12.5%)
Corpus	5 (11.11%)	7 (23.33%)	7 (25%)	1 (12.5%)	1 (12.5%)
Corpus + tail	2 (4.44%)	3 (10%)	3 (10.71%)	3 (37.5%)	1 (12.5%)
Tail	7 (15.56%)	18 (60%)	10 (35.71%)	3 (37.5%)	0
Head + corpus + tail	1 (2.22%)	0	0	0	0

IPMN—intraductal papillary mucinous neoplasm, MCN—mucinous cystic neoplasm, SCA—serous cystadenoma, CPEN—cystic pancreatic endocrine neoplasm, SPN—solid pseudopapillary neoplasm.

**Table 5 medicina-59-00241-t005:** Types of procedures in specific types of pancreatic cystic tumors (PCTs).

	PCT Type					
	IPMN (*n* = 45)	MCN (*n* = 30)	SCN (*n* = 28)	SPN (*n* = 8)	CPEN (*n* = 8)	All
Pancreatoduodenectomy	20 (44.44%)	0	6 (21.43%)	4 (50%)	1 (12.5%)	31
*Traverso*	17 (37.78%)	0	6 (21.43%)	4 (50%)	1 (12.5%)	28
*Whipple*	2 (4.44%)	0	0	0	0	2
*Clagett*	1 (2.22%)	0	0	0	0	1
Distal pancreatectomy with splenectomy	3 (6.67%)	13 (43.33%)	10 (35.71%)	0	1 (12.5%)	30
Extended distal pancreatectomy with splenectomy	2 (4.44%)	9 (30%)	7 (25%)	0	4 (50%)	22
Extended distal pancreatectomy	6 (13.33%)	2 (6.67%)	1 (3.57%)	1 (12.5%)	1 (12.5%)	12
Distal pancreatectomy	2 (4.44%)	4 (13.33%)	1 (3.57%)	0	0	7
Total resection	5 (11.11%)	0	0	0	0	5
Subtotal resection	4 (8.89%)	0	0	0	0	5
Central pancreatectomy	1 (2.22%)	0	2 (7.14%)	2 (25%)	0	5
Enucleation	0	1 (3.33%)	1 (3.57%)	0	1 (12.5%)	3
Others	2 (4.44%)	1 (3.33%)	0	1 (12.5%)	0	4

IPMN—intraductal papillary mucinous neoplasm, MCN—mucinous cystic neoplasm, SCN—serous cystic neoplasm, CPEN—cystic pancreatic neuroendocrine tumor, SPN—solid pseudopapillary neoplasm.

**Table 6 medicina-59-00241-t006:** Types of postoperative complications following surgery for pancreatic cystic tumors (PCTs).

Postoperative Complications Following Surgery for Pancreatic Cystic Tumors		
Type of Complication	Number of Complications	Types of Surgical Procedures
Overall postoperative morbidity	43 (34.68%)	25/43 DP 13/43 PD 2/43 CP 2/43 TP 1/43 Enucleation
Intra-abdominal fluid collection	21 (16.94%)	11/21 DP 6/21 PD 3/21 TP 1 Enucleation
Pancreatic postoperative fistula (POPF)	18 (14.52%)	15/18 DP 2/18 PD 1/18 CP
Biliary fistula	8 (6.45%)	5/8 PD 2/8 TP 1/8 DP
Acute pancreatitis	4 (3.2%	3/4 DP 1/4 CP
Wound infection	5 (4.03%)	2/5 DP 2/5 PD 1/5 CP
Dehiscence of gastrojejunal anastomosis	1 (0.8%)	1/1 PD
Dehiscence of duodenojejunal anastomosis	1 (0.8%)	1/1 PD
Intra-abdominal hemorrhage	1 (0.8%)	1/1 DP
Pulmonary embolism	1 (0.8%)	1/1 DP
Reoperation rate	15 (12.1%)	8/15 DP 4/15 PD 3/15 TP
Mortality	2 (1.61%)	1/2 CP 1/2 TP

DP, distal pancreatectomy; PD, pancreaticoduodenectomy; CP, central pancreatectomy; TP, total pancreatectomy.

## Data Availability

The data presented in this study are available on request from the corresponding author. The data are not publicly available as the data were collected from patients who were hospitalized in the Department of Digestive Tract Surgery.

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
