# Peer review of "Pancreatic Cystic Tumors: A Single-Center Observational Study"

_medicina, 2023, doi:10.3390/medicina59020241_

Round 1

Reviewer 1 Report

COMMENTS TO THE AUTHOR(S)
This is a review of ID: medicina-2061284, “Pancreatic cystic tumors: a single-center observational study.”. This manuscript is very interesting, but there are a few points that need to be revised.

Major Comments:

There is an abundance of previous reports on the characteristics of each pancreatic cystic lesion. There is little significance in examining a small number of cases. The discussion should be short and to the point. The Discussion section is somewhat redundant, and discussion of facts based on previous reports should be shortened, and active consideration of current clinical issues should be encouraged.

Minor comments

1)     Introduction line 48,49

The literature should be cited as to the basis for the ratio of each pancreatic cystic lesion.

2)     Introduction line 74-76

It is a unique citation, but it can also be deemed unnecessary for academic consideration.

3)     Introduction

How about mentioning invasive diagnostics such as cystic fluid analysis by EUS-FNA and CLE for cystic lesions of the pancreas? These are still research topics, and it is expected that the content could emphasize the importance of diagnostic imaging in clinical practice.

4)     Results

Please unify "CPEN" and "CPNET" with one another.

5)     Results line200-208

It is inferred that there would not be a strong need to discuss the results of PD and DP in detail. It may be worthwhile to mention here that there was no difference in PD or DP outcomes, or in complications, depending on the type of pancreatic cystic lesion.

6)     Discussion

The discussion is too long. Since the main focus of this manuscript is a single-center, retrospective study, not a review, how about a literature discussion of the current clinical issues based on data from your own institution?

Author Response

Dear Editor,

Dear Reviewer,

 Thank you for peer reviewing of our manuscript  medicina- 2061284, entitled "Pancreatic cystic tumors: a single-center observational study”.

Thank you for your questions and comments. We have fully addressed all the comments and my responses appear below. Our revised work includes corrections according to reviewers’ comments in the text. The changes, made according to reviewers’ comments, are marked using the „Track Changes” function in the main manuscript.

We take this opportunity to express my gratitude to the reviewers for their constructive and useful remarks. Their comments allowed us to identify areas in my manuscript that needed modification.

We also thank you for allowing me to resubmit a revised copy of the manuscript.

We hope that the revised manuscript is now acceptable for publication in Medicina.

Responses to Reviewer 1.

COMMENTS TO THE AUTHOR(S)

Comment:

This is a review of ID: medicina-2061284, “Pancreatic cystic tumors: a single-center observational study.”. This manuscript is very interesting, but there are a few points that need to be revised.

Answer:

Thank you for your positive feedback. The manuscript has been revision according to all your suggestions.

Major Comments:

Comment:

There is an abundance of previous reports on the characteristics of each pancreatic cystic lesion. There is little significance in examining a small number of cases. The discussion should be short and to the point. The Discussion section is somewhat redundant, and discussion of facts based on previous reports should be shortened, and active consideration of current clinical issues should be encouraged.

Answer:

Thank you for this comment. There are previous reports on the characteristics of each single pancreatic cystic lesion in the world literature, but there are not many reports including summary and comparative analysis of all five types of PCTs. Our report includes and discusses all five types of PCTs. A small number of cases in our analysis is associated with a fact that our study is retrospective and single-center. A single-center nature of our study limits an impact of confounding factors related to differences in the surgical technique and clinical experience for surgical outcomes in patients received surgery in different centers. Although our study includes a smaller cohort, it is a Polish / Central European voice in the discussion on this subject. There are only a few papers related to this subject and our study is a Polish / Central European voice in the worldwide discussion. Moreover, our results may be used in a further meta-analysis. So far, there are not many meta-analyses regarding summary analysis of different types of PCTs.

Minor comments

Comment:

1)     Introduction line 48,49

The literature should be cited as to the basis for the ratio of each pancreatic cystic lesion.

Answer:

The literature has been cited as follows:

Pancreatic cysts are divided into pseudocysts (75%) and true cysts (25%) [1]. Among true cysts, we distinguish neoplasmatic (10%), congestive (10%) and congenital (5%) cysts. Cystic pancreatic neoplasms account for 1–5% of all primary pancreatic tumors [1].

 Comment:

2)     Introduction line 74-76

It is a unique citation, but it can also be deemed unnecessary for academic consideration.

Answer:

This citation has been removed according your suggestion.

Comment:

3)     Introduction

How about mentioning invasive diagnostics such as cystic fluid analysis by EUS-FNA and CLE for cystic lesions of the pancreas? These are still research topics, and it is expected that the content could emphasize the importance of diagnostic imaging in clinical practice.

Answer:

A paragraph regarding invasive diagnostics such as cystic fluid analysis by EUS-FNA and CLE for cystic lesions of the pancreas has been added in the introduction as follows:

Currently, clinical assessment, radiological investigations (computed tomography (CT) and magnetic resonance imaging (MRI) or magnetic resonance cholangiopancreatography (MRCP) as well as endoscopic ultrasonography (EUS) with fluid analysis,(fluid cytology, carcinoembryonic antigen (CEA) level and biopsy of the cystic wall) are used in standard diagnostic process in patients with PCTs, According to the literature, cytological investigation of the cyst fluid can differentiate mucinous from non-mucinous PCTs with 42% sensitivity and 99% specificity, while cyst fluid CEA ≥ 192 ng/mL can differentiate mucinous from non-mucinous cyst with 52-78% sensivity and 63-91% specifity. Cyst amylase level <250 U/L (4.2 μkat/L) can exclude the presence of a pseudocyst with 44% sensitivity and 98% specificity. Fluid cytology and CEA level are not useful investigations in differentiation of mucinous PCTs as MCN and IPMN. EUS-guided needle-based confocal laser scanning endomicroscopy (CLE) is useful for visualization and assessment of the pancreatic cyst epithelial wall [11]. This investigation, coupled with intravascularly injected fluorescein dye, shows the vascular PCT pattern revealing distinct epithelial features [12]. Needle-based CLE used with EUS-based fine-needle aspiration and/or biopsy might improve diagnostic accuracy in order to reduce unnecessary surgery and patient observation [11]. A multidisciplinary approach with a combination of analysis of medical history and clinical presentation as well as above mentioned investigations as cross-sectional radiological imaging (CT, MRI, MRCP), EUS with cyst fluid cytology, cyst fluid analysis and cyst wall biopsy with CLE and molecular profiling is used in differential diagnostics and risk stratification of PCTs [12].

Comment:

4)     Results

Please unify "CPEN" and "CPNET" with one another.

Answer:

According your suggestion, „CPEN” and „CPNET” have been unified and „CPEN” is used in the whole manuscript, tables and figures.

 Comment:

5)     Results line200-208

It is inferred that there would not be a strong need to discuss the results of PD and DP in detail. It may be worthwhile to mention here that there was no difference in PD or DP outcomes, or in complications, depending on the type of pancreatic cystic lesion.

Answer:

According to your suggestion, details regarding the results of PD and DP have been limited and it has been mentioned that there was no difference in PD or DP outcomes as well as in complications, depending on the type of PCT as follows:

3.2. Surgical details and postoperative complications

The most frequent surgery among IPMN was pancreatoduodenectomy (PD) (44.4%), while among MCN and SCN – distal pancreatectomy (DP) (81%). The type of surgery depended on tumor location. There was no difference in PD or DP outcomes as well as in complications, depending on the type of PCT. There was association between surgical procedure and operation time (p<0.0001). Post-surgery hospitalization time was longer in patients following PD (Figure 5). The complete presentation of surgery types is visible in Table 5.

Comment:

6)     Discussion

The discussion is too long. Since the main focus of this manuscript is a single-center, retrospective study, not a review, how about a literature discussion of the current clinical issues based on data from your own institution?

Answer:

The discussion has been shortened and only comparison of our results with the literature data has been presented.

In discussion, we have presented characteristics of all types of PCTs and compared our data with the worldwide literature. In the second part, we compared our results regarding associations between PCTs and various parameters as well as our reports concerning surgical treatment ond outcomes with the other authors‘ reports.

Reviewer 2 Report

It is a well-structured paper covering an interesting topic. There are a few minor comments to be addressed:

1 - Table 6 - will be great if you can provide another (third) column specifying what type of surgery was conducted and which one led to a particular postoperative side effect.

2 - Table 7 - perhaps will be better if you present this data in tex, rather than making a bulky table. Moreover, only one parameter deemed to be significant.

3 - Discussion is way too large. Please focus only on comparison of your study to the other available data. In addition, will be reasonable to discuss potential study limitations and future perspectives of your outcomes.

4 - A suggestion, rather than a comment - what makes you consider that your study is the largest report from Poland? I may be wrong, but Johns Hopkins have a collaboration with Warsaw clinics and I can recall a study from they centre with almost 2000 patients recruited and analysed. In addition, conclusion should provide some insight into what is the future and promising (if there are any) results for further patients care.

Otherwise, the manuscript is well prepared and I am happy to read an amended version later. 

Author Response

Dear Editor,

Dear Reviewer,

Thank you for peer reviewing of our manuscript  medicina- 2061284, entitled "Pancreatic cystic tumors: a single-center observational study”.

Thank you for your questions and comments. We have fully addressed all the comments and my responses appear below. Our revised work includes corrections according to reviewers’ comments in the text. The changes, made according to reviewers’ comments, are marked using the „Track Changes” function in the main manuscript.

We take this opportunity to express my gratitude to the reviewers for their constructive and useful remarks. Their comments allowed us to identify areas in my manuscript that needed modification.

We also thank you for allowing me to resubmit a revised copy of the manuscript.

We hope that the revised manuscript is now acceptable for publication in Medicina.

Responses to Reviewer 2.

Comment:

It is a well-structured paper covering an interesting topic. There are a few minor comments to be addressed:

Answer:

Thank you for your positive feedback. The manuscript has been revision according to all your suggestions.

Comment:

1 - Table 6 - will be great if you can provide another (third) column specifying what type of surgery was conducted and which one led to a particular postoperative side effect.

Answer:

According your suggestion, Table 6 has been extended and another (third) column specifying what type of surgery was conducted and which one led to a particular postoperative side effect as follows:

Table 6. Types of postoperative complications following surgery for pancreatic cystic tumors (PCTs).

Postoperative complications following surgery for pancreatic cystic tumors

Type of complication

Number of complications

Types of surgical procedures

Overall postoperative morbidity

43 (34.68%)

25/43 DP

13/43 PD

2/43 CP

2/43 TP

1/43 Enucleation

Intra-abdominal fluid collection

21 (16.94%)

11/21 DP

6/21 PD

3/21 TP

1 Enucleation

Pancreatic postoperative fistula (POPF)

18 (14.52%)

15/18 DP

2/18 PD

1/18 CP

Biliary fistula

8 (6.45%)

5/8 PD

2/8 TP

1/8 DP

Acute pancreatitis

4 (3.2%

3/4 DP

1/4 CP

Wound infection

5 (4.03%)

2/5 DP

2/5 PD

1/5 CP

Dehiscence of gastro-jejunal anastomosis

1 (0.8%)

1/1 PD

Dehiscence of duodeno-jejunal anastomosis

1 (0.8%)

1/1 PD

Intra-abdominal hemorrhage

1 (0.8%)

1/1 DP

Pulmonary embolism

1 (0.8%)

1/1 DP

Reoperation rate

15 (12.1%)

8/15 DP

4/15 PD

3/15 TP

Mortality

2 (1.61%)

1/2 CP

1/2 TP

DP, Distal pancreatectomy; PD, Pancreaticoduodenectomy; CP, Central pancreatectomy; TP, Total pancreatectomy.

Comment:

2 - Table 7 - perhaps will be better if you present this data in tex, rather than making a bulky table. Moreover, only one parameter deemed to be significant.

Answer:

According your suggestion, Table 7 has been removed and this data has been presented in the text as follows:

Postoperative morbidity and mortality was 34.68% and 1.66%, respectively. Intra-abdominal fluid collection and clinical relevant postoperative pancreatic fistula (POPF) were the most frequent complications. All postoperative complications are presented in Table 6. There was association between American Society of Anaesthesiologists’ (ASA) classification and postoperative complication rate. A higher number of complications was noted in patients with higher ASA score (Figure 7). Spearman test showed significant positive correlation between ASA score and hospitalization time in patients undergoing DP (body+tail) with splenectomy (R=0.614; p=0.00308).  There was no significant correlation between ASA score and hospitalization time in patients undergoing DP (tail) with splenectomy, PD (Traverso), and DP (body + tail).There was no significant correlation between body mass index (BMI) in patients undergoing all above mentioned procedures.

Comment:

3 - Discussion is way too large. Please focus only on comparison of your study to the other available data. In addition, will be reasonable to discuss potential study limitations and future perspectives of your outcomes.

Answer:

The discussion has been shortened and only comparison of our results with the literature data has been presented. The discussion has been shortened and only comparison of our resuls with the literature data has been presented.

In discussion, we have presented characteristics of all types of PCTs and compared our data with the worldwide literature. In the second part, we compared our results regarding associations between PCTs and various parameters as well as our reports concerning surgical treatment ond outcomes with the other authors‘ reports.

Regarding discussion concerning potential study limitations and future perspectives of our outcomes, in a primary version of our manuscript, a paragraph presenting above mentioned points had been presented at the end of discussion (just above conclusions). In the revised manuscript, it has been extended as follows:

The limitation of this study is the single-center and retrospective analysis as well as a small number of CPEN (n=8) and SPT (n=8) groups. A single-center nature of our study limits an impact of confounding factors related to differences in the surgical technique and clinical experience for surgical outcomes in patients received surgery in different centers. Although our study includes a smaller cohort, it is a Polish / Central European voice in the discussion on this subject.

A prospective, multi-center study is needed to show the epidemiology and treatment results of PCTs in Poland. Despite a retrospective and single-center design of our study, its results can be used in a large meta-analysis regarding epidemiology and surgical outcomes in patient with PCTs. It is also possible to compare these results with the other worldwide literature data regarding PCTs.

Comment:

4 - A suggestion, rather than a comment - what makes you consider that your study is the largest report from Poland? I may be wrong, but Johns Hopkins have a collaboration with Warsaw clinics and I can recall a study from they centre with almost 2000 patients recruited and analysed. In addition, conclusion should provide some insight into what is the future and promising (if there are any) results for further patients care.

Answer:

Thank you for this comment. We have reviewed single-center reports on PCTs from Polish centers and we have found the largest number of 145 pancreatic cysts including 48 PCTs (Wlaźlak et al., Łódź, Polish Journal of Surgery 2018) and 46 PCTs in our previous report (Jabłońska et al., Katowice, Polish Journal of Surgery 2017). We have not found larger analysis in the worldwide literature from Poland. Therefore, it has been written: „In conclusion, this is the first and largest one-center report from Poland for resected PCTs”. I may be wrong, so I will be grateful for the exact references for this work with Warsaw clinics participation. In addition, we can remove this sentence from our manuscript if you wish.

According your suggestion, conclusions have been extended in order to provide some insight into what is the future and promising (if there are any) results for further patients care as follows:

  1. Conclusions

In conclusion, this is the first and largest one-center report from Poland for resected PCTs. Female predominance and distal pancreatic location in most PCTs (besides IPMN with male and pancreatic head predominance), young age, and the benign nature of most PCNs were noted. IPMNs were the smallest while MCNs were the largest tumors in our patients. Malignant transformation and metastases to lymph nodes was reported in patients with CPENs, IPMNs, and MCNs. One distant metastasis was reported in a patient with MCN. There was no metastasis in patients with SCNs and SPNs. According to our analysis, the distribution of different types of PCTs is similar to American and other European populations and differs from Eastern (Indian and Chinese) publications.

Due to substantial postoperative morbidity and a risk of malignancy in patients undergoing surgery for PCTs, a proper selection of candidates for surgical treatment is very important and challenging. Management in patients with PCTs is still very challenging due to difficulties in differentiation non-malignant and malignant neoplasms, which determines the proper decision. The most important is the proper selection of patients requiring surgery at the right time, without unnecessarily exposing patients who do not require surgical treatment to complications related to surgery. Our study as well as the other reports have been showed that pancreatic resection performed even in high-volume surgical centers is associated with a relative significant risk of postoperative morbidity despite of progression in surgical techniques. The correct algorithm of surveillance of patients not qualified for resection is also important. Knowledge of epidemiology and clinical presentation of specific types of PCTs is as important for clinicians as modern radiological cross-sectional investigations in differential diagnostics and making decision regarding management of PCT. Therefore, many further retrospective and prospective, single- and multicenter studies, as well as meta-analyses based on their results, are needed in order to modify and precise current guidelines for the management of PCTs.

Comment:

Otherwise, the manuscript is well prepared and I am happy to read an amended version later. 

Answer:

Thank you for your positive opinion. I am happy to improve our article according to your valuable suggestions. I hope you will be satisfied our revision.

Round 2

Reviewer 1 Report

COMMENTS TO THE AUTHOR(S)
This is a review of ID: medicina-2061284, “Pancreatic cystic tumors: a single-center observational study.”.

The manuscript has been improved according to reviewer's comments, and has great content.

Please make further revisions in the following points.

Minor comments

1)      Please unify "SPT" and "SPN" with one another.

Author Response

Dear Editor,

Dear Reviewer,

 Thank you for peer reviewing of our manuscript  medicina- 2061284, entitled "Pancreatic cystic tumors: a single-center observational study”.

Thank you for your questions and comments. We have fully addressed all the comments and my responses appear below. Our revised work includes corrections according to reviewers’ comments in the text. The changes, made according to reviewers’ comments, are marked using the „Track Changes” function in the main manuscript.

We take this opportunity to express my gratitude to the reviewers for their constructive and useful remarks. Their comments allowed us to identify areas in my manuscript that needed modification.

We also thank you for allowing me to resubmit a revised copy of the manuscript.

We hope that the revised manuscript is now acceptable for publication in Medicina.

Responses to Reviewer 1.

COMMENTS TO THE AUTHOR(S)

Comment:

The manuscript has been improved according to reviewer's comments, and has great content.

Answer:

Thank you for your positive opinion. The manuscript has been revised according to all your suggestions.

Comment:

1)      Please unify "SPT" and "SPN" with one another.

Answer:

According your suggestion, „SPT” and „SPN” have been unified and „SPN” is used in the whole manuscript, tables and figures.
